# Preparation and Thermoelectric Properties of Semiconducting Single-Walled Carbon Nanotubes/Regioregular Poly(3-dodecylthiophene) Composite Films

**DOI:** 10.3390/polym12112720

**Published:** 2020-11-17

**Authors:** Mengdi Wang, Qin Yao, Sanyin Qu, Yanling Chen, Hui Li, Lidong Chen

**Affiliations:** 1State Key Laboratory of High Performance Ceramics and Superfine Microstructure, Chinese Academy of Sciences, Shanghai 200050, China; wangmd@shanghaitech.edu.cn (M.W.); qusanyin@mail.sic.ac.cn (S.Q.); chenyl2@shanghaitech.edu.cn (Y.C.); lihui889@mail.sic.ac.cn (H.L.); cld@mail.sic.ac.cn (L.C.); 2China School of Physical Science and Technology, Shanghai Tech University, Shanghai 201210, China; 3Center of Materials Science and Optoelectronics Engineering, University of Chinese Academy of Sciences, Beijing 100049, China

**Keywords:** thermoelectric, semiconducting SWNT, poly(3-dodecylthiophene), composite

## Abstract

Single-walled carbon nanotubes (SWNTs) have been widely used as leading additives for improving the thermoelectric properties of organic materials, due to their unique structure and excellent electronic transport properties. However, the as-synthesized SWNTs are mixtures (mix-SWNT) of semiconducting (sc-SWNT) and metallic (met-SWNT) carbon nanotubes. The significantly different surface character and transport behavior of sc-SWNT and met-SWNT frequently raise the difficulty of modifying microstructures, and tuning transport properties of SWNTs/organic composites, when using mix-SWNTs as dispersion phase. Herein, we prepared high quality sc-SWNTs/rr-P3DDT composite film by presorting pure sc-SWNT from the raw mix-SWNTs using regioregular poly(3-dodecylthiophene) (rr-P3DDT). Both the smoothness and compactness of sc-SWNTs/rr-P3DDT are great improved, as compared with the mix-SWNTs/rr-P3DDT films, and the sc-SWNTs are well-dispersed and uniformly wrapped by rr-P3DDT with diameter less than 50 nm. The significantly enhanced Seebeck coefficients and power factors are obtained in the sc-SWNT/rr-P3DDT samples. As the result, the maximum power factor of 60 μW/mK^2^ in 50 wt% sc-SWNTs sample is 70% higher than that of mix-SWNTs/P3DDT sample. This work reveals the effectiveness of pure semiconductor SWNTs as fillers to optimize the thermoelectric properties of CNT/polymer nano-composites.

## 1. Introduction

Organic thermoelectric materials, such as polyaniline [1,2], poly(3-hexylthiophene) (P3HT) [3] and poly(3,4-ethylenedioxythiophene) (PEDOT) [4,5,6], have attracted extensive attention as promising thermoelectric materials due to their low cost, flexibility and inherently low thermal conductivity. However, as compared with the state-of-the-art inorganic thermoelectric (TE)materials, the dimensionless figure of merit ZT (ZT = σS^2^T/κ, where σ, S, and κ are the electrical conductivity, Seebeck coefficient, and thermal conductivity, respectively) of organic compounds is much smaller because of their lower power factor (S^2^σ).

The nanocomposite approach has been utilized to tune the thermoelectric properties of conducting polymers based on the expectation of combining unique characters of individual nano-inclusions and/or introducing possible synergistic effects. It has been proven by quite a few studies that filling single-walled carbon nanotubes (SWNTs) into the polymer matrix is an effective way to enhance the thermoelectric power factor. This is generally due to the introduction of one-dimensional structures, with excellent electrical conductivity and hybrid structures, formed on the interfaces between SWNTs and polymer molecules [7,8]. Generally, there are two typical carbon nano-tubes, semiconducting and metallic ones, and as-produced SWNTs by all known synthetic methods contain both species [9]. The different surface characteristics and transport behavior of the two different types of SWNTs frequently raise the difficulty of modifying microstructures and tuning transport properties of SWNTs/organic composites when using the mixture SWNTs (mix-SWNTs) as the dispersion phase [10,11]. Recently, sc-SWNTs have been used to control the chirality distribution and carrier density of conducting polymers, yielding a higher power factor [12], compared to the use of mixed-SWNTs filler. However, few studies have been reported on the TE properties of sc-SWNT/polymer composites due to the arduous sorting of sc-SWNTs out of as-grown mix-SWNTs.

Recently, it has been demonstrated that sc-SWNT species can be selectively dispersed in conjugated polymer-containing solution and then be sorted from the raw mixture SWNTs [13,14]. Among various conjugated polymers, poly(thiophene), and its derivatives, have been found to most powerfully disperse and separate SWNTs because of their suitable diameter and chirality [15,16,17]. Lee et al. successfully sorted high purity sc-SWNTs from mix-SWNTs by using regioregular poly(3-alkylthiophene)s (rr-P3ATs) contained toluene solution, in which the distinctive side chains of rr-P3ATs make them selectively wrap up the sc-SWNTs [15]. On the other hand, poly(thiophene) and its derivatives (e.g., P3HT) have also been recognized as promising thermoelectric materials because of their appropriate energy gap and wide-doping ranges [3,18].

Here we successfully obtained pure sc-SWNTs powders by sorting them from raw mix-SWNTs through a simple solution centrifugation process using rr-P3DDT as the selective dispersant. The sc-SWNT/rr-P3DDT composite films with different sc-SWNTs contents were further fabricated by directly casting the sc-SWNTs and P3DDT contained solution. The sc-SWNTs are highly uniformly dispersed in rr-P3DDT polymer matrix. The thermoelectric power factor of Fe(TFSI)_3_-doped sc-SWNTs/rr-P3DDT composite film with 50% SWNT content reaches 60 μW /mK^2^, which is 70% higher than that of mix-SWNT/rr-P3DDT composite film and 12 times as that of pure rr-P3DDT film.

## 2. Materials and Methods

### 2.1. Raw Materials

The raw SWNTs were purchased from Timesnano Company of China (Shenzhen, China). rr-P3ATs with different side chains were purchased from Sigma Company (Shanghai, China). (CF_3_SO_2_)_2_NH (95%) was purchased from Shanghai Aladdin company (Shanghai, China). Fe_2_(SO4)_3_, NaHCO_3_, and toluene were provided by Sinopharm Chemical Reagent Company (Shanghai, China).

### 2.2. Sorting sc-SWNTs from Raw Mix-SWNTs

The raw SWNTs (5 mg) and the rr-P3ATs with different side chains (10 mg), were suspended in toluene (25 mL). This suspension was kept in a temperature-controlled cooling bath and sonicated for 30 min. The solution was subsequently centrifuged for 150 min at 42,000 *g* to remove SWNT bundles and insoluble components. The supernatant containing polymer-wrapped sc-SWCNTs and excess polymer is collected via pipette. The sc-SWNTs were then precipitated from the supernatant by long-time cooling in ice bath. And then the sc-SWNTs powder was obtained by removing residue polymer through subsequent filtering, toluene-rinsing, drying (at 45 °C) process.

### 2.3. Fabrication of sc-SWNT/P3DDT Composite Films

The sc-SWNTs were dispersed in heated P3DDT-contained toluene solution (1 mol/L P3DDT) in different contents. The intrinsic composite films were prepared by dropping the mixed solution on glass substrates and subsequently drying. The film was further doped by immersing it into dopant Fetf contained nitromethane (Fe(TFSI)_3_) solution, which was pre-synthesized by treating freshly prepared Fe(OH)_3_ (using (CF_3_SO_2_)_2_NH acid) in anhydrous nitromethane [3]. After doping treatment for 1.5 h, the doped composite films were rinsed with anhydrous nitromethane and dried under vacuum at 40 °C. The sc-SWNTs contents were identified by the initial weight ratio of the sc-SWNTs and the process-ended composite film. The obtained products were denoted as sc-SWNT/P3DDT. As a comparison, the doped composite films using raw mix-SWNTs with different contents were also prepared through the same process and the obtained products were denoted as mix-SWNT/P3DDT.

### 2.4. Characterization of Microstructure and Electrical Transport Properties

The morphology of films was probed by scanning electron microscopy (Supra45, Zeiss, Jena, Germany) (SEM), atomic force microscope (MPF-3D, Asylum Research, Santa Barbara, CA, USA) (AFM) and transmission electron microscopy (JEM-2100F, JEOL, Tokyo, Japan) (TEM). Raman spectra measurements were performed using a Raman spectrometer (JY LabRam-1B, HORIBA, Paris, France, λ_exc_ = 785 nm). The measurement of electrical conductivity and Seebeck coefficient along the in-plane direction were performed by a four-point probe method with ZEM-5 System (Advance-RIKO, Osaka, Japan). The thickness of the films was measured using a profilometer (Bruker, Hamburg, Germany).

## 3. Results

### 3.1. Process of Preparing sc-SWNT/rr-P3DDT Composite Films

The rr-P3AT, with different side chains, were utilized to sort sc-SWNTs from the raw mix-SWNTs using a simple solution process reported previously [15,17], as shown in Figure 1a. The resonant Raman scattering spectra (in the region of radial breathing modes, RBM) of the filtered toluene-based supernatants containing various kinds of rr-P3ATs are shown in Figure 2. The chirality index of SWNTs peaks are marked in the figure based on the position of the RBM peaks and Kataura plot [19,20]. There are no any obvious peaks of met- or sc-SWNTs for the supernatants of rr-P3HT-SWNT and rr-P3DT-SWNT solutions, indicating that neither met-SWNTs nor sc-SWNTs are captured by rr-P3HT or rr-P3DT. On the other hand, one strong peak of sc-SWNTs (10, 2) is observed for rr-P3OT-SWNT supernatant, and three strong peaks of sc-SWNTs ((12, 1), (8, 6) and (10, 2)) are observed for rr-P3DDT-SWNT supernatants. The met-SWNTs related peaks are not observed in either rr-P3OT-SWNT or rr-P3DDT-SWNT supernatants. Furthermore, the content of sc-SWNTs in rr-P3DDT-SWNT supernatant is higher than that in rr-P3OT supernatant. Based on the above Raman spectra results, it can be concluded that, among all polythiophene derivatives, the rr-P3DDT has the most powerful ability to stably and selectively suspend sc-SWNTs, and therefore, is suitable for sorting sc-SWNTs from the mix-SWNTs. It is observed that the carbon nanotubes are wrapped by P3DDT molecules forming supramolecular structure, and the P3DDT-wrapped sc-SWNTs are kept suspended in supernatant and able to precipitate in the subsequent cooling process. The side-chain length, conformation and density of side chains are all important parameters that govern the ability for conjugated polymers to form the desired supramolecular structures with certain SWNTs [15,16]. Due to the well-matched structures and surface charge characters between P3DDT molecular (side-chain length, conformation, density of side chains, dopant distribution) and sc-SWNTs (arrangement of carbon rings, doping state and defect distribution on the surface), the desired supramolecular structures were formed on the surface of carbon nanotubes.

### 3.2. Microstructure Analysis of sc-SWNT/rr-P3DDT Composite Films

The sc-SWNTs/P3DDT composite films were prepared by mixing the collected sc-SWNTs powders (Figure 1a) with P3DDT in toluene and casting the solution on the glass substrate, as shown in Figure 1b. As comparison, the mix-SWNTs/P3DDT composite films were also fabricated using the same process. The surface scanning electron microscopy (SEM) images of the as-prepared composite films with different SWNT contents were shown in Figure 3. The surface of mix-SWNT/P3DDT film is observed porous and rough, with a visible morphology of CNT bundles in diameter about 100~200 nm exposed on the surface (Figure 3a–c). On the other hand, the surface of sc-SWNT/P3DDT film is smooth and compact, and the polymer-coated carbon nanotubes with diameter about 50 nm are clearly observed (Figure 3a’–c’).

We further characterized the freeze-fractured cross-sectional surfaces of the mix-SWNT/P3DDT and sc-SWNT/P3DDT composite films with 50 wt% SWNTs by SEM and AFM (atomic force microscope), and the images are shown in Figure 4. The cross-sectional SEM image shows that the mix-SWNT/P3DDT composite film is loose, with large quantity of carbon nanotubes pulled out from the matrix (Figure 4a), while the carbon nanotubes are cross-linked to form uniform three-dimensional (3D) network structure in the sc-P3DDT composite film (Figure 4a’). Moreover, the average roughness (Ra) of mix-SWNT/ P3DDT film is up to 13 nm, while the Ra of sc-SWNT/P3DDT is only 4 nm, as shown in Figure 4b,b’. The great morphology diversity of mix-SWNT/P3DDT and sc-SWNT/P3DDT composite films could be ascribed to the different dispersion states of carbon nanotubes in toluene solvent. The SWNT/P3DDT wrapping morphology was further checked by TEM on the toluene-dissolved mix-SWNT/P3DDT and sc-SWNT/P3DDT composite films (in SWNTs content of 50 wt%), as shown in Figure 4b, b’. It was observed that the SWNTs aggregate seriously with large quantity of bundles more than 200 nm in diameter for the mix-SWNT/P3DDT sample, While, the P3DDT-wrapped single SWNTs with diameter less than 50 nm are uniformly dispersed for the sc-SWNT/P3DDT sample. The P3DDT wrapping on single nanotube of sc-SWNT not only enable selective suspending of sc-SWNT in P3DDT/toluene solvent (as shown in the sorting process) but also promotes dispersion of sc-SWNTs in the solvent forming non-aggregated, smooth and compact sc-SWNT/P3DDT composite films.

### 3.3. Thermoelectric Properties of sc-SWNT/rr-P3DDT Composite Films

The electrical conductivity and Seebeck coefficient of pure P3DDT, sc-SWNT film and mix-SWNT film are listed in Table 1. The electrical conductivity of mix-SWNT film is up to 1610 S/cm, obviously higher than that of pure sc-SWNT film because met-SWNTs generally show higher electrical conductivity than sc-SWNTs [11,12]. Based on the measured density and the measured electrical conductivity of the films, the electrical conductivity of a single carbon nanotube is estimated as 2690 S/cm and 741 S/cm for mix-SWNT and sc-SWNT, respectively. The measured Seebeck coefficients of sc-SWNTs and mix-SWNTs are 12 μV/K, and 9 μV/K, respectively, much lower than pure P3DDT film.

The measured electrical conductivity and Seebeck coefficient of mix-SWNT/P3DDT and sc-SWNT/P3DDT composite films with different SWNT content, at room temperature, are shown in Figure 5. As shown in Figure 5a, the measured electrical conductivity for both sc-SWNT/P3DDT and mix-SWNT/P3DDT composite films increase with the increase of the SWNTs content. Since SWNT and P3DDT have similar densities, about 1 g cm^−3^, the volume fraction of SWNTs and P3DDT can be substituted by the weight fraction. The calculated conductivities are obtained based on parallel structure model ($\sigma =\sigma_{c}x+\sigma_{p}\left(1-x\right)$, where x is the volume fraction of the SWNTs in the composite, and the subscripts c and p denote the component carbon nanotube, and component polymer P3DDT, respectively. The estimated values of the electrical conductivity of the single carbon nanotubes and the pure doped-P3DDT listed in Table 1 are used as *σ_c_*, and *σ_p_*, respectively) [6] and percolation structure model of the mixture with two different conductive composites as following [21,22]:*σ*(*T*) = *σ*_0_(*T*) (*p* − *p_c_*)^u^ + *σ_L_*(*p* ≥ *p_c_*)(1)
*σ*(*T*) ≈ *σ_L_*(*p* < *p_c_*).(2)

Here, *σ* is the electrical conductivity of the final composite; *σ*_0_(*T*) is a fitted proportionality constant related to the intrinsic electrical conductivity of the filler SWNT; *p* is the volume fraction of the conducting phase in the composite; *p_c_* is the percolation threshold, which is the volume fraction at which the conductivity starts to increase; u is the value of the critical exponent. As shown in Figure 5a, it is found that the electrical conductivity of both mix-SWNTs/P3DDT composite films and sc-SWNTs/P3DDT composite films increases with increase of SWNTs content, and the experimental values are a little bit higher than the calculated values of mixture model. However, between those of the mixture and percolation models, which suggests that enhancement of electrical conductivity can be reasonably explained by the mixture rule of parallel model with partial percolation structure, caused by the addition of high conductivity carbon nanotubes into polymer matrix with low conductivity. Turning to the Seebeck coefficient, it is worth noting that, the Seebeck coefficients of sc-SWNTs/P3DDT composite films are significantly higher than those of mix-SWNTs/P3DDT composite film as shown in Figure 5b, though the Seebeck coefficient of pure sc-SWNTs is close to that of pure mix-SWNTs and much lower than that of pure P3DDT film (Table 1). Furthermore, the measured Seebeck coefficients of sc-SWNTs/P3DDT composite films are much large than the calculated values ($S=\frac{S_{c}\sigma_{c}x+S_{p}\sigma_{p}\left(1-x\right)}{\sigma_{c}x+\sigma_{p}\left(1-x\right)}$) based on the mixed model, while that of the mix-SWNT/P3DDT films is quite close to the calculation. The Seebeck coefficient of sc-SWNTs/P3DDT composite film with 50% SWNT content are about 30 μV/K, twice of the mix-SWNTs/P3DDT composite film. This difference is caused by the different dispersion and different SWNT/organic interfacial structure. Previous studies have reported that energy filtering effect can improve the Seebeck coefficient in inorganic/polymer hybrids by which interfaces form energy barriers that preferentially scattered low-energy carriers [23,24]. there are larger number of nano interfaces in sc-SWNTs/P3DDT composite formed by the highly uniform dispersion of sc-SWNTs in P3DDT. Furthermore, the P3DDT-wrapped SWNT formed supramolecular structure due to the hybrid bonding between SWNTs and P3DDT. Both of these two factors may contribute to the enhancement of Seebeck coefficients due to the strong energy filtering effect in the SWNTs/P3DDT hybrid interface. The maximum power factor of up to 60 μW/mK^2^ is obtained for 50% sc-SWNT/P3DDT film, which is 70% higher than that of mix-SWNTs/P3DDT composite film with the same SWNT content.

## 4. Conclusions

The selective behavior of regioregular poly(3-alkylthiophene)s (rr-P3ATs) with various side chains to disperse different kinds of SWNTs was investigated, and it was found that, among all rr-P3ATs derivatives, P3DDT possesses the best ability to selectively suspend the semiconducting SWNTs in toluene solution. Then, the high purity sc-SWNTs were successfully sorted from raw mix-SWNTs through a simple solution centrifugation process using P3DDT as the dispersant, and the sc-SWNTs/P3DDT composite films were fabricated by directly casting the mixed solution of sc-SWNTs and P3DDT on the glass substrate. The P3DDT molecules is more inclined to wrap on the surface of sc-SWNTs, which not only enables the molecule to selectively capture and sort the sc-SWNTs from the mix-SWNTs in the toluene solution, but also prevent the aggregation of carbon nanotubes in the solution. The sc-SWNTs uniformly dispersed in the rr-P3DDT polymer matrix with the diameter less than 50 nm, forming smooth and compact composite films. The measured electrical conductivity of SWNTs/P3DDT composite films increases with increasing the SWNT content, and can be reasonably explained by the mixture rule of parallel model with partial percolation structure. In particular, the sc-SWNTs/rr-P3DDT composite films exhibit much higher Seebeck coefficient than those of mix-SWNTs/P3DDT films, which may be ascribed to the strong energy filtering effect caused by the nano-interfaces between sc-SWNTs and rr-P3DDT. The maximum thermoelectric power factor (S^2^μ) of SWNT-S/PANI composite film with 50 wt% SWNTs reaches up to 60 μW/mK^2^, which is 70% higher than that of the mix-SWNTs/PANI composite film at the same composition. This work offers a facile and scalable route to prepare sc-SWNTs/polymer composites and to optimize the TE properties of the nanocomposites using sc-SWNTs as fillers.

## Figures and Tables

**Figure 1 polymers-12-02720-f001:**
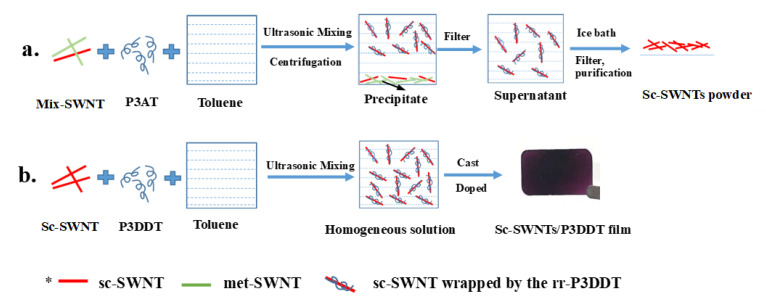
Illustration of the solution process of; (**a**) sorting sc-SWNTs from raw mix-SWNTs; and (**b**) preparing sc-SWNT/P3DDT composite film.

**Figure 2 polymers-12-02720-f002:**
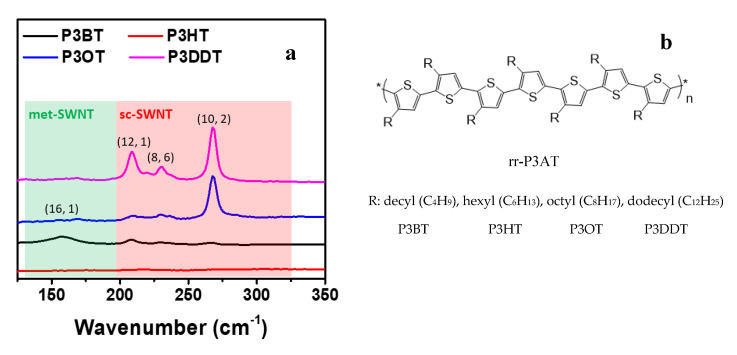
(**a**) RBM regions of resonant Raman scattering spectra at 1.58 eV (785 nm) excitation energies of the supernatants containing sc-SWNTs with various rr-P3ATs as dispersant; (**b**) Chemical structures of the various polythiophenes investigated.

**Figure 3 polymers-12-02720-f003:**
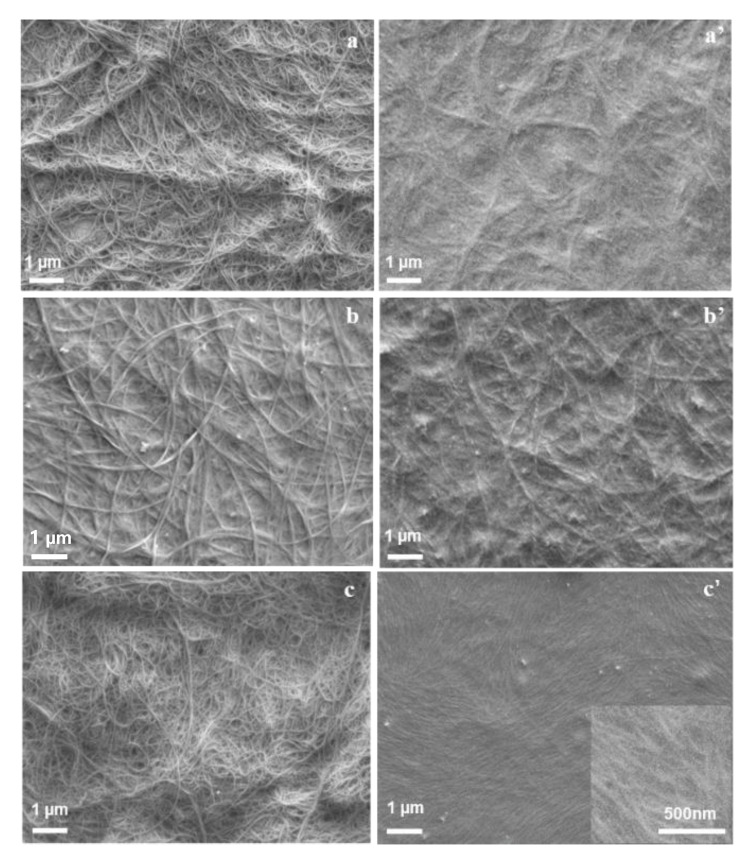
The surface SEM images of mix-SWNT/rr-P3DDT composite films (**a**–**c**) and sc-SWNT/rr-P3DDT composite films (**a’**–**c’**) with 20 wt%, 33 wt% and 50 wt% SWNT content.

**Figure 4 polymers-12-02720-f004:**
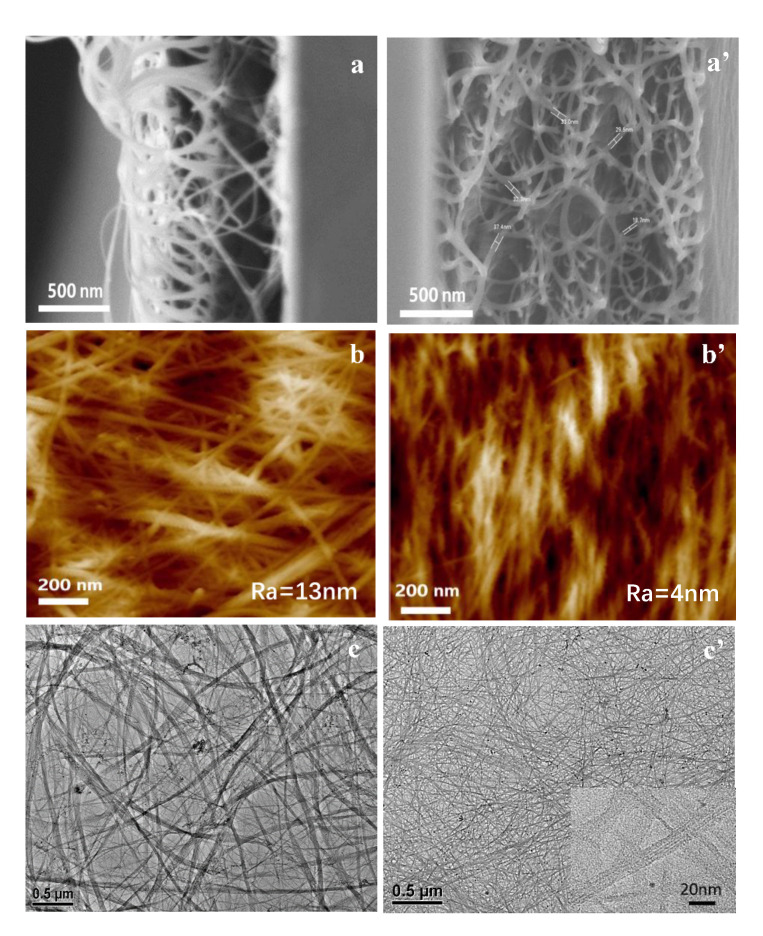
The freeze-fractured cross section SEM images and AFM topographic images of mix-SWNTs/ rr-P3DDT composite films (**a**,**b**) and sc-SWNTs/ rr-P3DDT composite films (**a’**,**b’**) with 50 wt% SWNTs; TEM images of SWNTs/ rr-P3DDT composite films; (**c**) and sc-SWNTs/ rr-P3DDT composite film (**c’**) with 50 wt% SWNTs dissolved in toluene solvent.

**Figure 5 polymers-12-02720-f005:**
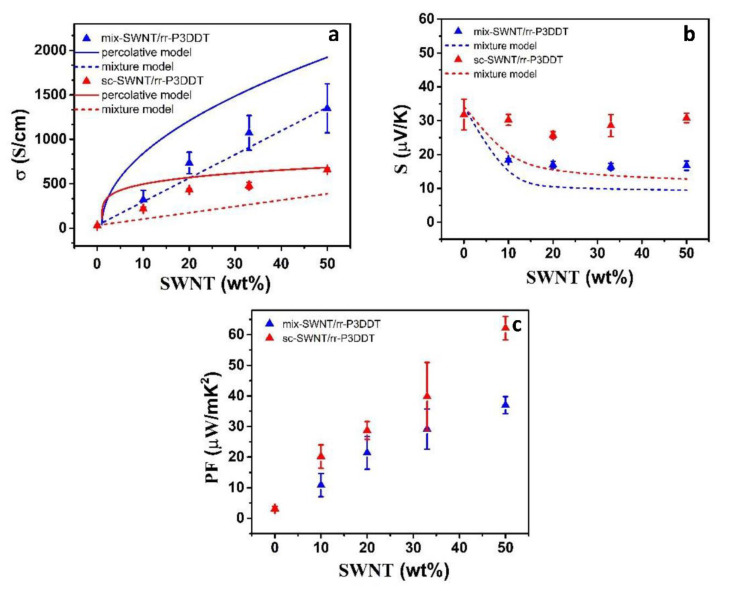
The triangles are experimental values of; (**a**) electrical conductivity; (**b**) Seebeck coefficient; and (**c**) power factor for mix-SWNTs/P3DDT composite films (blue) and sc-SWNTs/P3DDT composite films (red) with different SWNTs content. The dash line and solid line are the calculated values based on mixture model, and percolation model, respectively.

**Table 1 polymers-12-02720-t001:** Electrical conductivity and Seebeck coefficient of pure P3DDT film, sc-SWNT film and mix-SWNT film.

Sample	Electrical Conductivity (S cm^−1^)	Seebeck Coefficient (μV K^−1^)
pure P3DDT film	33	34
pure sc-SWNT film	741	12
pure mix-SWNT film	2690	9

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
