# Peer review of "Preparation and Thermoelectric Properties of Semiconducting Single-Walled Carbon Nanotubes/Regioregular Poly(3-dodecylthiophene) Composite Films"

_polymers, 2020, doi:10.3390/polym12112720_

Round 1

Reviewer 1 Report

The manuscript "Preparation and thermoelectric properties of semiconducting SWNT/rr-P3DDT composite films" shows a relatively simple method to obtain the combination of SWCNT and poly (3-dodecyl thiophene) to obtain composites.

The manuscript well written with interesting results and there only few points to address.

Regarding the title please don,t use abbreviation there as rr-P3DDT is not used in general for poly (3-dodecyl thiophene). Please write full name.

It also would be helpful in introduction to give more evidence of poly (3-dodecyl thiophene)how it benefits and why as example don,t use PEDOT:PSS instead.

Please include mean values with standard deviation in your results not showing in Figure 5. Regarding Figure 5b why the mixture model so different from the experimental model?

Reviewer 2 Report

The author of the paper has shown many experimental results, but some answers are required.

1. There is no picture of how composites are created as the length of the alkyl chain changes.

2. The longer the length of the alkyl chain, the better the mixing with the CNT, should be explained.

3. The separation of CNTs does not make much difference thermoelectric performance. There should be an explanation for this.

4. There should be more specific explanations for thermoelectric performance improvements resulting from the formation of composite materials with CNT.

Round 2

Reviewer 2 Report

Through the kind and adequate responses of authors, the doubts are dispelled and the errors of the manuscript are correctly revised. I think the authors provide quite reasonable real time sensor detect experiment with real-time changes.

Thus, I gladly recommend that this manuscript might be suitable to publish to this journal.